# Buffalo on the Edge: Factors Affecting Historical Distribution and Restoration of *Bison bison* in the Western Cordillera, North America

**Jonathan James Farr** [1,*] **and Clifford A. White** [2]

1   Department of Biological Sciences, University of Alberta, Edmonton, AB T6G 2E9, Canada
2   CW and Associates, 1018 14 St., Canmore, AB T1W 1V5, Canada
*   Correspondence: jfarr@ualberta.ca

**Abstract:** The historic western edge of the bison (*Bison bison*) range and the ecological processes that caused its formation are frequently debated with important implications for bison restoration across North America. We test the hypothesis that a combination of bottom-up habitat suitability and top-down harvest pressure from Indigenous peoples were important processes in forming the western edge of bison distribution. Using 9384 historical journal observations from 1691–1928, we employ MaxEnt ecological niche modelling to identify suitable bison habitat across the Western Cordillera from bottom-up climatic, land cover, and topographic factors. We then use mixed-effect logistic regression to test if bison occurrence in journal records can be in part explained by the abundance of humans, wolves, or grizzly bears, in addition to MaxEnt-derived habitat suitability. We find support for our hypothesis because of the limited suitable habitat in the Rocky Mountains that likely prevented westward bison dispersal from their core habitat, and there was a negative relationship between bison occurrence and human harvest pressure. On this basis, we propose that intensive human harvest from large populations in the Western Cordillera, subsidized by other wildlife, salmon, and vegetation resources, is an underappreciated socioecological process that needs to be restored alongside bison populations. Co-managing bison with Indigenous peoples will also mitigate the adverse effects of overabundant bison populations and maximize the ecological and cultural benefits of bison restoration.

**Keywords:** bison; restoration; socio-ecological processes; indigenous harvest; maximum entropy modelling



## 1. Introduction

In the period from ~CE 1750 to ~CE 1880, the Eurasian colonization of western North America caused one of the greatest near extinctions ever documented. Within a century, *Bison bison*, commonly called buffalo and once numbering in the millions across the grasslands and woodlands of the Great Plains, were reduced to fewer than 1000 animals [1]. Humans' role in the overkilling of bison is well-documented [2], and researchers mapped the spatial pattern of extirpation [3,4] within years of its occurrence (Figure 1). The political ramifications of this slaughter of the continent's largest land mammal helped stimulate conservation by the national governments of both the United States and Canada, including designating wildlife as a public resource [5,6] and withholding massive areas as public lands mandated for the sustainable use of wildlife and plant resources [7–9].

Today, the parks, protected areas, and public lands of North America's Western Cordillera, stretching from the Rocky Mountains westward to the Pacific from Yellowstone to Yukon, provide one of the greatest opportunities for the restoration and conservation of bison and many other species. Yellowstone in the United States and Banff in Canada are the birthplaces of the world's first national park systems, and both have played roles in the initial efforts to save the bison from extinction. Moreover, these parks are the cores of an area that now constitutes one of the planet's largest areas of public lands—a network

that provides ecological connectivity along the Cordillera from Yellowstone to Yukon [10]. These parks and other public lands are also the homelands of Indigenous peoples and provide ecosystem services, natural resources, and recreational opportunities for millions of local and regional residents. Furthermore, they are heavily visited by domestic and international travellers, providing an opportunity to observe, hunt, and study in the wilds of world-renowned scenery.

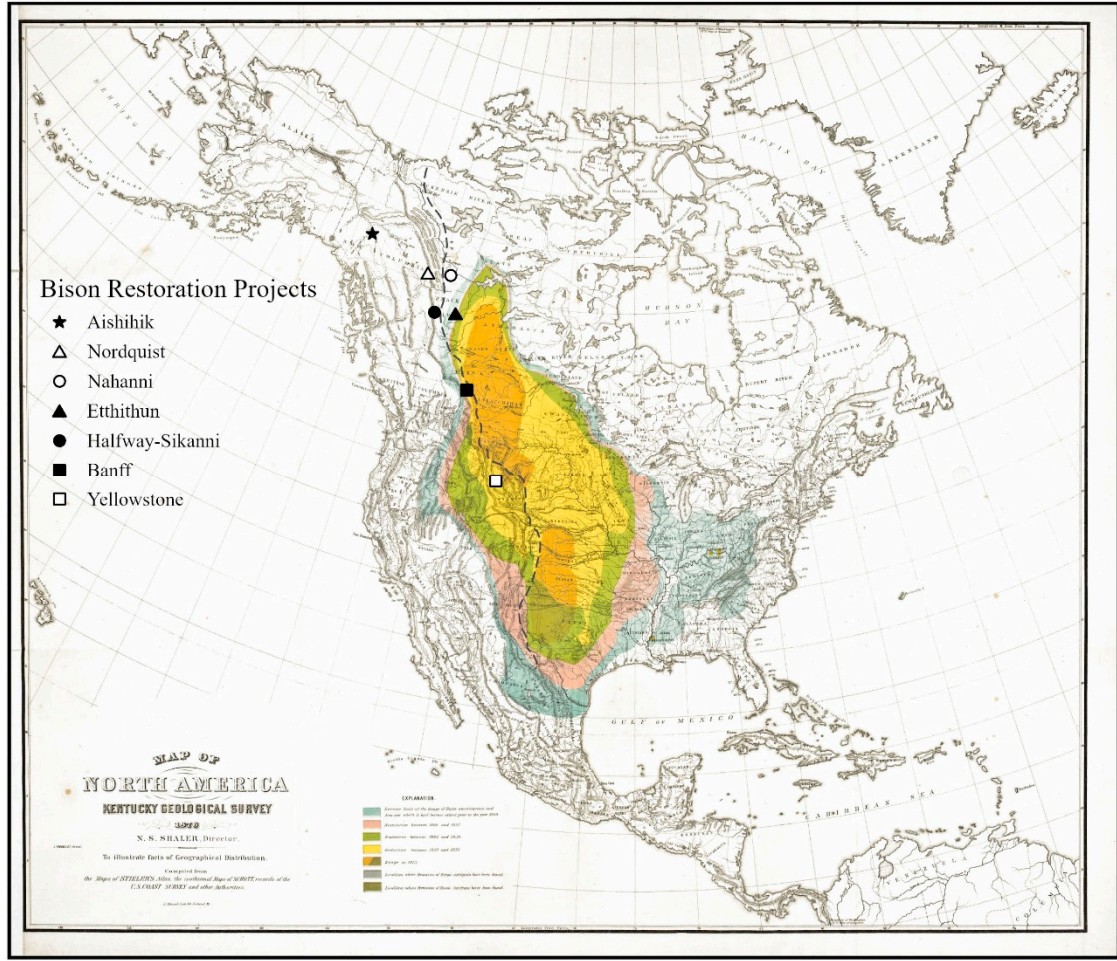

**Figure 1.** Range contractions of the American bison during the period from pre-1800 to 1875 as mapped by Joel Allen of the Kentucky Geological Survey in 1876 [3], showing the locations of the Western Cordillera restoration projects described in this paper. The figure shows the extreme edge of the historic bison range before 1800 (blue), the range contraction from 1800 to 1825 (pink), 1825 to 1850 (green), 1850 to 1875 (yellow), and the range in 1875 (orange). The dotted line delineates where the mountains of the Western Cordillera meet the Great Plains.

However, many of the parks and protected areas of the Western Cordillera are located on the historic edge of core bison distribution, and bison restoration in this area requires the consideration of the factors creating this dynamic. These factors are still debated, and several researchers have provided hypotheses for the scarcity of bison west of the Rocky Mountains [11–14], including lower forage quality, heavy winter snows, discontinuous habitat, and high harvest-caused mortality from large populations of Indigenous peoples. These explanations are not mutually exclusive and have likely interacted to cause low-density bison populations in this area. Empirical support for these hypotheses is limited; however, one recent study used archeological data and bioclimatic models to conclude that during the late Holocene, a source–sink dynamic likely existed between abundant bison populations in the Great Plains and limited populations in the Western Cordillera [13].

Although they did not explicitly examine the role of Indigenous hunting, these authors proposed that both marginal habitat and hunting pressure may have created this dynamic.

There is extensive evidence of intensive human bison harvest in the Rocky Mountains [14–17], and, if this process affected bison abundance in the Western Cordillera, then bison restoration projects need to consider the effects of two interacting phenomena. First, there is the potential historic socio-ecological system resulting from the human involvement in creating low-density areas where bison abundance was limited. In these areas with historically no or very few bison, vegetation and other ecosystem attributes may be potentially ill-adapted to high numbers of large and gregarious herbivores [18]. The second phenomenon is socio-political. The early establishment and management of national parks and other public lands followed a socially and politically driven tradition of the removal of Indigenous and other human influences [19,20], and in parks, ongoing programs of "natural regulation" were established to allow wildlife populations to rise and fall unimpeded by human influences [21,22]. These decisions effectively eliminated any socio-ecological roles of Indigenous peoples. In situations where harvest might have maintained low bison densities, creating few ecological impacts, the recent and relatively novel perspective of modern management may allow high numbers of bison, elk, and other herbivores that greatly impact ecosystems [23–26].

In this paper, we further inform bison restoration by examining the role that the human harvest of bison may have historically played in creating socio-ecological systems that limited bison populations in the Western Cordillera. Using a large database of historic journal observations from northwest North America, we empirically test the hypothesis that bison populations in the Western Cordillera were limited by both low bottom-up habitat suitability and high top-down human harvest. First, we present historic information on bison and their potential primary predators [27–29]: humans, wolves (*Canis lupus*), and grizzly bears (*Ursus arctos horribilis*), using abundance and distribution data from a biome-scale record of the first-person journals of early non-Indigenous travellers. Second, we conduct analyses of these data to test how bottom-up habitat suitability based on MaxEnt modelling and mixed-effect modelling of top-down processes, namely wolf and bear predation and human harvest, could have limited bison occurrence in the Western Cordillera. Third, we review the demographics and range areas of bison restored on the edge or outside of the historic range to further test whether the growth of these populations is limited by their location. We then discuss the restoration of bison in accordance with bottom-up and top-down ecological processes and long-term human influences.

## 2. Materials and Methods

### 2.1. First-Person Journal Observations

Our analysis required specific observations of bison locations and abundance, human cultural practices, and the abundance of other predator species present in historic landscapes. Historic bison, human, wolf, and grizzly bear occurrence and abundance were indexed using the first-person daily wildlife observations obtained from the journals of European mariners, fur traders, trappers, and government mappers for the periods of historical range contraction (Figure 1): CE 1691–1860 in southern Canada and the northern United States, and CE 1770–1928 in northern Canada and eastern Alaska.. Our study area encompassed the northwestern portion of historic bison distribution in North America (Figure 2). While this excluded a large portion of the historic bison range in the southwest and east portions of North America, we chose to focus on restoration initiatives within the Western Cordillera and thus delineated our study area accordingly. We collected journal observations from the Pacific Ocean and the Alaska–Yukon border in the west to Lake Winnipeg and the Hudson Bay in the east, and from the Arctic Ocean in the north to approximatively the state boundaries of California, Nevada, Utah, and Colorado in the south.

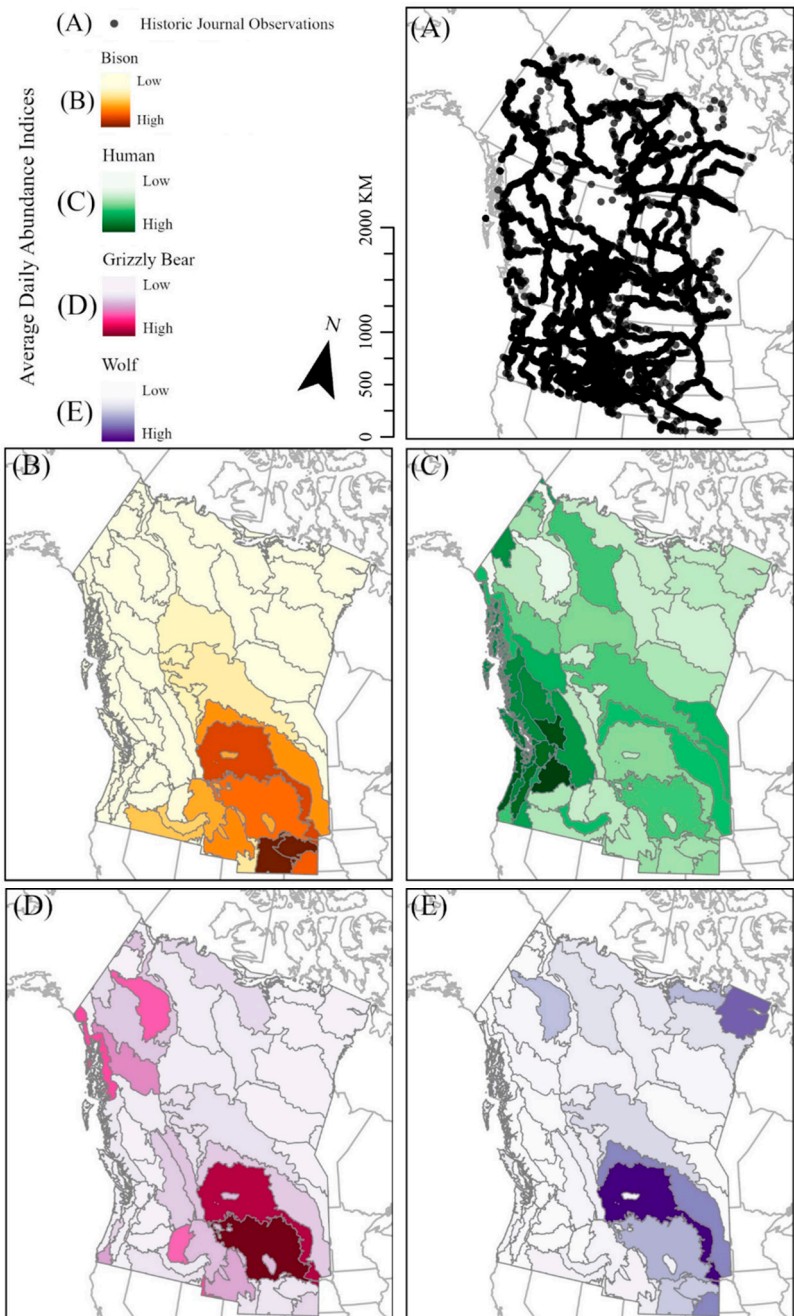

**Figure 2.** Study area boundary and location of historic journal observations from CE 1691–1860 in southern Canada and northern United States, and CE 1770–1928 in northern Canada and eastern Alaska (**A**). The mean abundance indices per ecoregion of bison (**B**), humans (**C**), grizzly bears (**D**) and wolves (**E**). Abundance was indexed following the methods of Kay [30], based on signs, harvest, and group size.

Our methods followed Kay's procedures for tallying observations from the Lewis and Clark journals [30]. For bison, wolves, and bears, three measures quantify the observations of journalists. The first is the animals seen, where a value of 1 was assigned if journalists reported an old sign, 2 if the sign was fresh, and 3 if they actually saw the animal. The second index is animals killed, where either the exact number killed was recorded, or where "some" or "a few" was recorded as 3, "several" as 7, and "many" as 10. The third measure is the herd or group size. When journalists report sighting large numbers of a species, a value of 10 was assigned, and 5 was assigned for moderate amounts. The animals seen and killed, and the herd/group size were then added together to obtain a measure of

abundance. Observations made by journalists at long-term camps or trading posts may be tallied by specified periods of 4 to 30 days with the total kill numbers for the period. For quantifying human abundance, if an old sign was observed, this was assigned a 1, a fresh sign a 2, and if the journalists actually saw people a 3 was assigned, or 10 if the human group size was greater than ten. Further, the quality of the journal observation was rated as "ND" or no data for the day/period, or low, moderate, or high depending on the level of detail. The location was plotted as the nightly campsite, and again from low to high quality depending on the journalist's description of the location. For all analyses, we excluded observations where the wildlife or location data quality was rated as no data. The complete database for these observations (in spreadsheet and Google Earth format) is available in the Supplementary Data.

To visualize the large-scale trends in bison, human, wolf, and grizzly bear abundance based on historic journal observations, we averaged the abundance indices for the North American ecoregions [31,32] mapped for our study area. We used an ecoregion scale because they are useful for delineating terrestrial biodiversity patterns for global land-use planning and conservation across taxa [33] and have recently been used in studies of large mammal restoration [34]. Where ecoregion boundaries extend beyond the study area, the mean resource index includes observations from across the ecoregion.

### 2.2. Journal Data Analysis

#### 2.2.1. Bottom-Up Effects on Bison

We first accounted for the effect of bottom-up factors on bison distribution by constructing an ecological niche model for bison using maximum entropy modelling (MaxEnt) [35]. We predicted that the distribution of bison would be largely explained by bottom-up modelling but expected that the addition of top-down factors (see below) would also predict bison occurrence and thus provide evidence of their importance in defining the historic bison range. MaxEnt has been widely used to estimate habitat suitability and predict the geographic distribution of a variety of species [36], including both European and North American bison [13,37]. The output of MaxEnt modelling provides an estimated probability of species occurrence based on environmental variables, and we join others in referring to this as synonymous with habitat suitability [36,38,39].

To represent the bottom-up factors that could affect habitat suitability, we collected a series of land cover, topographic, and climatic predictors in ArcGIS Pro version 2.4.0 [40]. The temporal span during which land cover and climatic predictors were measured does not reach back to time periods when bison were observed and reported by journalists, and land cover and climate have both changed significantly over recent centuries [41]. Our analysis assumes that these variables have remained static over time, and, in part to account for this assumption, we applied current data measured at a large scale (raster pixel size = 200 square kilometers) to describe the general land cover and climatic characteristics of locations where bison were reported. Furthermore, we included a covariate representing the human modification of terrestrial systems to account for changes in land cover since historical times (e.g., the development of cities) [42,43].

For the land cover variables, we reclassified the MODIS 2005 land cover map into grassy and treed areas at a 250 m cell size [44] and calculated their proportional areas within each pixel. To capture the topographic variation, we calculated the terrain ruggedness using the USDS North America Elevation 1-km resolution grid [45]. We calculated a curvature grid in ArcGIS and then used a moving window analysis to determine the standard deviation of curvature within a 3 km radius of each 1-km cell to produce a measure of topographic variability, with higher values indicating more rugged terrain [46]. We then averaged the ruggedness values within each pixel. Lastly, we collected five climatic variables from a global climate model at a 1 km pixel size, which were quantified from the climate normal between 1961 and 1990 [47]. We considered the annual precipitation as snow and the mean temperature of the coldest month to be indicative of snow depth and winter severity, which have been previously shown to affect bison distribution [48,49].

The other three climatic variables were the mean annual solar radiation, the mean summer precipitation, and the mean temperature of the warmest month, which we considered indicative of primary productivity and thus forage availability for bison. Again, we caution that our analysis assumes that modern climates are representative of historic trends, and given the occurrence of the Little Ice Age during the journal observation period, this may affect our conclusions as climates have warmed substantially.

We randomly split bison journal observations ($N$ = 1654) into training (75%) and testing (25%) and used the kuenm package in R [50] to calibrate 17 candidate models using the *kuenm_cal* function. We selected the best models with *kuenm_ceval*, and then created the final models using the full set of occurrences and selected parameters using *kuenm_mod*. We created the candidate models including all possible combinations of environmental variables, 14 values of regularization multiplier (0.1–1.0 at intervals of 0.1, 2–5 at intervals of 1, 8 and 10), and linear, quadratic, and product feature classes. We excluded hinge or threshold feature classes in an attempt to minimize overfitting by limiting the model complexity [51]. We evaluated the models based on statistical significance (partial ROC) [52], omission rates (E = 5%) [53], and model complexity (AICc $\leq$ 2 from top model) [54]. To evaluate the relative contributions of our environmental predictors, we determined the percentage contribution of each variable toward the model fit [13] and generated marginal response curves to evaluate the effect of these variables on bison habitat suitability. Finally, we generated 10 replicates of the single top-selected model using k-fold cross-validation [50], averaged their logistic outputs, binned the values into 10 equal interval categories, and projected this across our study area to quantify bison habitat suitability on a scale from 1 (low suitability) to 10 (high suitability).

### 2.2.2. Top-Down Effects on Bison

To test the hypothesis that top-down processes acted in addition to bottom-up habitat suitability to limit bison distribution in the Western Cordillera, we modelled the relationship between the presence of bison and the abundance of three species that consume bison (humans, wolves, and grizzly bears) using mixed-effect logistic regression [55]. If the bottom-up factors were adequate at explaining historic bison distribution, we would expect that the top-down factors would fail to significantly explain any variation in bison distribution. We assumed that the abundances of humans, wolves, and grizzly bears were representative of the harvest or predation intensity that they posed towards bison at each location.

For each journal observation for which data were available, we rated the bison presence as 1 if any evidence of bison was observed, and 0 if not. We then used the *glmer* function from package lme4 [56] to model the presence/absence as a function of habitat suitability as determined by the MaxEnt logistic output value at the observation location, and the observed abundance index values for humans, wolves, and grizzly bears. All predictors were mean centered and scaled to a standard deviation of 1, and we verified that no predictors were highly correlated ($r^2$ > 0.6). We also included the journal observer as a random effect to account for spatial and temporal autocorrelation within journal expeditions. We then used the *dredge* function from package MuMIn to identify the top model(s) based on the most parsimonious combination of predictor variables ($\Delta AIC_c$ < 2 from the top model) [57]. We evaluated the model fit of our top-selected model(s) based on conditional and marginal $R^2$ values using the *r2* function from the package performance [58]. We conducted all statistical analyses in Rstudio version 2022.2.3.492 [59,60] and considered the results significant if the *p* values were <0.05.

### 2.3. Cordilleran Bison Restoration Project Demographics and Outcomes

Finally, we tested whether the historic edge of the current bison range has limited the success of current restoration projects by summarizing the range area and demographics (current population estimates, carrying capacity, growth rates) of bison in populations outside of the historic core distribution. We predicted that the current outcomes of these projects may provide further insight into the processes delineating historic bison distribution.

## 3. Results

### 3.1. Historical Range and Abundance of Bison

Of the 9384 journal observations where wildlife and locational data were robust, bison were reported 1841 times. Bison were mostly reported as sightings (81.2%), but other evidence of their presence was also recorded (18.4%) based on traditional ecological knowledge acquired from guides or other Indigenous people, feces, tracks, or wallows. Humans were reported as seen or encountered in 3272 journal observations, and in another 438 records, evidence of humans was observed without encountering people. Evidence or visual sightings of wolves or grizzly bears were less frequently reported, in 183 and 243 journal observations, respectively.

Across ecoregions (Figure 2), bison abundance largely followed the proposed historic distribution (Figure 1) and was centered around the Great Plains ecoregions. Bison were rarely observed in the Rocky Mountains ecoregions and only extended west of the Rocky Mountains in the south end of the study area. In contrast, humans were most abundant west of the Rocky Mountains. The abundances of wolves and grizzly bears were reportedly highest in the same areas as bison, the prairie ecoregions, although both were also reported farther north in the region.

### 3.2. Factors Affecting Bison Distribution

As predicted, the bottom-up factors were strong predictors of bison occurrence based on MaxEnt modelling. All 17 candidate models were statistically better than the null model, and 16 of these met the omission rate criteria, but an examination of the model complexity revealed a single top model that included all the predictor variables (average area under the ROC curve = 0.864, omission rate at 5% = 0.048, AIC = 28,312.04). The predictor with the highest contribution to the model fit was the proportion of grass (43.5%), followed by the mean coldest month temperature (19.8%), the mean annual radiance (16.3%), human modification (6%), the proportional tree cover (5.6%), ruggedness (3.7%), the mean warmest month temperature (2.7%), precipitation as snow (1.3%) and the mean summer precipitation (1.2%). The marginal response curves (Appendix A, Figure A1) indicate that habitat suitability was positively related to the mean annual solar radiance and human modification level, negatively related to the mean summer precipitation, precipitation as snow, the mean warmest month temperature, and the proportional tree cover, and that intermediate values of the mean coldest month temperature, ruggedness, and proportion of grass yielded the highest habitat suitability values.

Habitat suitability varied across the study area, with the highest values occurring in the Great Plains east of the Rocky Mountains (Figure 3). The Rocky Mountains and higher elevation areas had much lower habitat suitability, as did areas in the far north and near the coasts. However, several areas in the Western Cordillera exhibited some habitat suitability, including the larger valleys of the Cordillera in Montana and Wyoming, the grassland interior plateau of British Columbia, and the plains of Idaho and Washington state.

Top-down factors were important predictors of bison occurrence, as a mixed-effect logistic regression analysis of bison presence revealed a clear top model (>5 AIC$_c$ from the next model) that included all predictor variables (Table 1). The conditional $R^2$ value was 0.520, while the marginal $R^2$ was 0.322, suggesting a reasonably good model fit. Bottom-up habitat suitability, quantified based on the MaxEnt output values, was the strongest predictor of bison presence. Of the top-down effects on bison distribution, only human abundance was negatively related to bison occurrence, while a positive relationship existed between the abundance of wolves and grizzly bears and bison presence (Table 1).

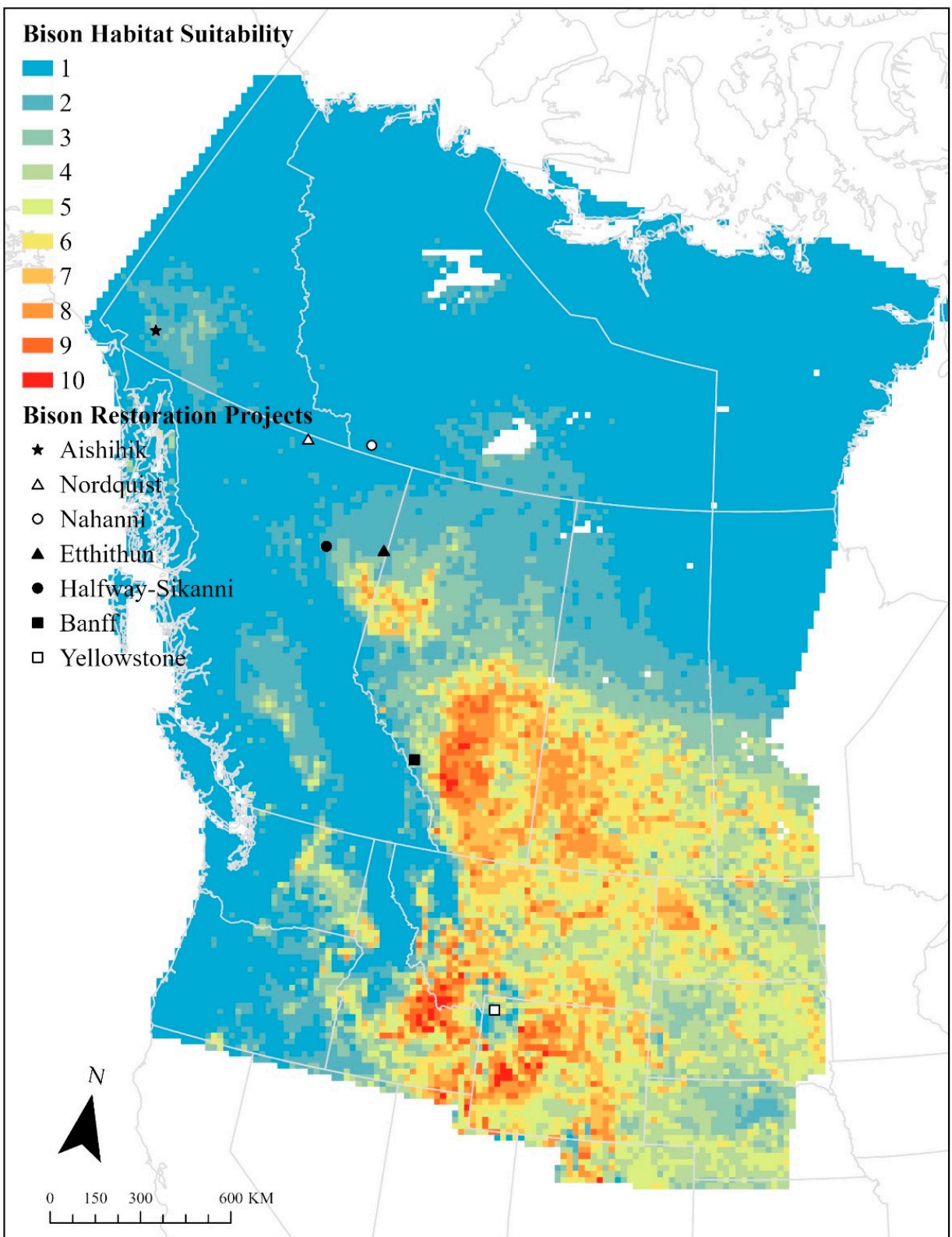

**Figure 3.** Bison habitat suitability (1 = low, 10 = high) as determined through MaxEnt modelling of bison occurrences based on historic journal observations. Models were constructed in R using the *kuenm* package and used climatic, land cover, and topographic variables as bottom-up predictors of bison occurrence.

**Table 1.** Top mixed-effect logistic regression model predicting bison presence or absence in historic journal records as a function of MaxEnt habitat suitability and the abundance of humans, wolves, and grizzly bears.

| Predictors | Estimate | Std. Error | *p* |
|---|---|---|---|
| (Intercept) | −2.76 | 0.192 | <0.001 |
| MaxEnt habitat suitability | 1.33 | 0.0575 | <0.001 |
| Human abundance index | −0.224 | 0.0404 | <0.001 |
| Grizzly bear abundance index | 0.0827 | 0.0291 | 0.0045 |
| Wolf abundance index | 0.201 | 0.0361 | <0.001 |

*3.3. Cordilleran Bison Restoration Project Outcomes*

　　Table 2 summarizes the demographics for seven ongoing bison restoration projects in the Cordillera. Although all native non-human predators occur in all areas, the annual population rates of increase are generally high (>10%) when the populations are <50% of the carrying capacity. Exceptions include projects such as Nordquist and Nahanni, which are bisected by highways, causing high bison mortality [61]. Generally, where the carrying capacity is estimated from the habitat quality, the estimated density is higher than for those where it is determined from social criteria. There is no apparent relationship between the demographics of the projects either on the edge or those outside the historic bison range. For example, although the Aishihik project in Yukon is several hundred kilometers NW of the historical bison range boundary, both the population growth rates and the potential population totals are comparable to those in or near the historic range.

**Table 2.** Demographics for select current bison restoration projects in the Western Cordillera, North America.

| Location and Ecotype [References] | Estimated Range (km²) | Population Estimate (Year) | Estimated Carrying Capacity (K, Bison/km²) | Observed Annual Rate of Increase (%) at ≤50% K | Remarks |
|---|---|---|---|---|---|
| **Yellowstone, WY, MT** **Plains Bison** [22,62,63] | 9400 | 4500 (2018) | 0.405 | 12 | All native predators except humans. K determined by habitat quality and demographics, population c. 2010 > 75% of K (counts of ~4000), large impacts on grasslands, woody plants, and hydrologic indicators, emigration rate out of park is increasing. |
| **Banff, AB** **Plains Bison** [64–66] | 1200 | 80 (2020) | 0.480 | 30 | All native predators except humans. K determined by habitat quality, current increase rate very high due to high female-to-male ratio of the founder herd, current population < 100 but increasing rapidly. |
| **Halfway-Sikanni, BC** **Plains Bison** [67–69] | 3200 | 1013 (2014) | 0.500 | 10–38 | All native predators present, including humans. A total of 48 plains bison escaped in 1971, population ~1300 in 2006, K socially determined to minimize range expansion. |

**Table 2.** *Cont.*

| Location and Ecotype [References] | Estimated Range (km²) | Population Estimate (Year) | Estimated Carrying Capacity (K, Bison/km²) | Observed Annual Rate of Increase (%) at ≤50% K | Remarks |
|---|---|---|---|---|---|
| **Etthithun, BC & AB Wood Bison** [61,68,70–72] | 5000 | 193 (2010) | No information | 15–20 | All native predators present. First Nations allocated permits to take bison to discourage range expansion due to conflicts with industry and agriculture. |
| **Nordquist, BC Wood Bison** [61,68,71,73–75] | 1400 | 124–142 (2010) | No information | <5% | All native predators present, including humans. A total of 49 bison released in 1995 at Aline Lake, by 2010, range mainly along 170 km Alaska Highway, population limited by traffic accidents. |
| **Nahanni, NT & BC Wood Bison** [76,77] | 11,700 | 431 (2011) | No information | ~8% | All native predators present, including humans. Slow growth since establishment, limiting factors appear to be drowning, traffic accidents, hunting, and perhaps tooth wear due to diet high in silica. |
| **Aishihik, YT Wood Bison** [78–82] | 11,000 | 1106–1385 (2011) | 0.125 | 10–20 | All native predators present. Socially determined carrying capacity to minimize conflicts with Indigenous land uses and competition with other species. |

## 4. Discussion

Our analysis of historical journal observations informs bison restoration in the Western Cordillera by supporting the hypothesis that top-down human harvest pressure, in addition to bottom-up habitat suitability, acted to influence bison population dynamics in this region. First, we found that bottom-up factors largely explained bison distribution and highlighted a large band of low-suitability habitat in the Rocky Mountains that isolated otherwise suitable areas in the Cordillera from the core bison range on the Great Plains. Second, we identified a negative relationship between bison occurrence and the abundance of humans even when bottom-up habitat suitability was included in models, suggesting that harvest pressure may have also been an important determinant of the western limit of bison distribution. The rapid expansion of bison populations restored to these areas of historically low habitat suitability provides further support for our hypothesis. Based on our findings, we propose that maximizing the ecological and cultural benefits of bison restoration may require the re-establishment of the role of Indigenous people as a key socio-ecological process to regulate bison populations in the Western Cordillera [83,84].

### 4.1. Processes Limiting Historic Bison Distribution

With a MaxEnt ecological niche modelling approach, we identified a large core area of highly suitable bison habitat from central Alberta, Canada, southward along the Rocky Mountains into the Great Plains, falling within the historic bison distribution mapped by Allen [3] and Roe [4] (Figures 1 and 3). Of the two recognized ecotypes of North American bison [85,86], the MaxEnt model largely encompasses the range of the migratory "plains" variant that occurred in high numbers on the grasslands of the southern Great Plains. Traditional ecological knowledge and numerous accounts of early journalists have provided evidence of plains bison herds numbering in the tens of thousands [30,87,88]. The range of "wood bison", a more sedentary ecotype that existed at low densities in the northern mixed and boreal forest, was identified as less suitable habitat by MaxEnt.

Within the core range, a combination of factors likely made bison abundant, including high forage availability from expansive grasslands [89,90], favorable climatic conditions [13], a lack of geographic barriers to formation, movements, and the dispersal of large herds [90,91], and areas that provided refuge from human harvest [89,91]. Indigenous groups in this area were heavily dependent on bison [2,92,93], but over-harvest was likely limited because bison found refuge both spatially, in intertribal buffer zones between conflicting Indigenous groups [30,94], and temporally during winter when large areas were mostly inaccessible to human hunters because of unavailable firewood and shelter on the exposed prairie [91,95,96]. Further north and west in forested areas, bison likely persisted despite lower habitat suitability due to spacing in low densities, low human populations, and high connectivity, leading to periodic dispersals from large bison herds to the south and east [30,97]. Source–sink wildlife population dynamics likely operated within this core area, both between and within plains and wood subspecies, to allow for persistent bison populations in many areas [30,88,98,99]. In contrast to the high numbers of bison found throughout the core distribution, few bison were reported west of the Rocky Mountains, and we suggest that this was caused by both more limited connected habitat and greater top-down harvest pressure from humans.

Outside of the core range, the model identified a large band of low-suitability habitat in the northern Rocky Mountains that likely limited bison dispersal. The topography of the foothills and mountains and deep winter snowpack also provided excellent opportunities for Indigenous people to harvest bison using natural traps and jumps. There is considerable archeological evidence of bison harvest in the northern Rockies [14–16,100,101], and this likely further limited bison dispersal westward. Connectivity was less of a barrier in the south, as habitat corridors existed in Wyoming and southwest Montana [102]. Indeed, low numbers of bison were historically observed west of the Rockies on the grasslands of southern Idaho, northern Utah, eastern Oregon, and southeastern Washington (Figure 2). Archaeological evidence also shows periodic bison presence in this region throughout the Holocene period [103,104]. Source–sink dynamics may have existed here, with low-density herds occasionally being replenished by larger herds dispersing from the core bison range on the Great Plains. During this period, bison body mass in these small herds declined in a pattern similar to those on the Great Plains, suggesting their origin was possibly from periodic dispersals through these corridors from the heartland of the bison habitat to the east [105]. Even without re-colonization from larger populations, some low-density populations may have been self-sustaining in the Western Cordillera despite low connectivity to the Great Plains populations. Future research should aim to characterize bison movement corridors [102], migration patterns [95], and source–sink population dynamics to better understand the historic distribution and abundance of bison populations.

Top-down harvest pressure from human populations was likely also higher both in the north and south portions of the Western Cordillera than in the core bison range, because they contained large human populations that were supported by substantial plant and salmon resources [106–108]. Our analysis and other research support the hypothesis that high human abundance is negatively related to bison occurrence due to the socio-political pressure to over-exploit bison populations where possible [30,88,98,99]. For example, the relatively recent extirpation of bison (<3000 BP) in Alaska and Yukon may be associated with increasing human numbers [109,110] and the development of hide trade routes from the Liard and Yukon Rivers to the Pacific coast [111,112]. The recent population growth of bison restored both at the edge and outside of their core historic range (Table 2) emphasizes that human harvest may have been as important, or potentially more limiting, than the bottom-up factors in these areas. We also emphasize that the effect of human harvest on bison interacted with various bottom-up factors that had more prevalent impacts in the Western Cordillera, such as deep snow, rugged topography with potential natural traps, and abundant wood resources from forests.

We acknowledge several limitations of our analysis that should be considered when interpreting our findings. First, as mentioned in the methods, the covariates used in MaxEnt

models assumed that the climatic and land cover variables measured in the 21st century are representative of historical patterns. While we believe that this assumption is appropriate for modelling broad-scale trends in bison distribution, changes in local climate and land cover [41] make more detailed interpretations of our habitat suitability model potentially perilous. In addition, climate warming since the period during which the observations were made may cause directional bias in our MaxEnt model to suggest bison habitat occurred more in cooler climates than was the case, shifting our model northward. Second, journal observations were largely located along historic travel routes that were influenced by climate, terrain, and food availability. This introduces another potential source of bias in this dataset towards river valleys and other low-resistant travel routes, which we attempted to account for by using a large (200 square kilometer) raster pixel size. Additionally, with the more intensive sampling in the southern part of our study area, our estimates may be biased towards this area. Third, historical journal observations are almost exclusively written from a male, colonial perspective. While the records of some journals describe traditional ecological knowledge shared by Indigenous guides or groups encountered when travelling, they often do not describe plant use or female perspectives on seasonal resource availability. Future research requires further Indigenous knowledge of historic bison and other resources to better understand the nuance of historic human–bison interactions [6,101,106,113–115].

*4.2. Implications for Bison Restoration in the Western Cordillera*

Despite these bottom-up and top-down limitations imposed on historical bison distribution, recent restoration projects have achieved rapid bison population growth in areas that were historically on the fringe or outside of what we identified as suitable habitat (Figure 3, Table 2). The success of these projects demonstrates that bison can thrive in a broad range of climates, vegetation conditions, and predators, even if snow depth [48], forage quality and availability [116], and non-human predation [117] influence their localized spatial distribution. Recent studies have also demonstrated remarkable plasticity in bison diets across North America [118,119], and MaxEnt ecological niche modelling based on fossil records of bison and historic climates demonstrated that much of North America has been suitable bison habitat over the past 20,000 years [13].

Bison can clearly thrive outside of their historical range with minimal human harvest pressure, but their potential adverse effects on areas that have not recently sustained high numbers of bison may present a challenge for bison restoration. These impacts can occur from intense herbivory on ill-adapted vegetation species [23–25,63,83], direct and apparent competition with other species [120], disease transmission [1,25], and detrimental cultural effects through human injury and damage to cultural resources [79,121]. However, restoring the top-down process posed by Indigenous peoples' harvest of bison may benefit bison populations from reduced disease prevalence in low-density herds under top-down pressure [122], higher social tolerance from more involved local communities [121,123], reduced risk of excessive herbivory that degrades their ecosystems [23,24,84], and, as an additional but substantial benefit, Indigenous people may restore deep cultural connections with bison [124]. Our analysis supports the hypothesis that this top-down process may have limited bison populations in the Western Cordillera, adding to a large body of recent evidence demonstrating that Indigenous people were a key ecological force shaping North American landscapes by affecting the abundance of large mammals [30,88], cultivating and gathering plant and animal species [114,125], and manipulating fire regimes across the continent [113,126–128]. Restoring these socio-ecological dynamics may help restore bison across the Western Cordillera.

Of the bison restoration projects discussed in this paper, Indigenous harvest as a socio-ecological process has been partially restored in Aishihik, Nahanni, Nordquist, Etthithun, and Halfway-Sikanni. In the national parks of Banff and Yellowstone, Indigenous harvest has not yet been restored, and this top-down process remains de-coupled from these bison populations. Free-roaming bison have only recently been restored to Banff in 2018 [64,65], and implementing Indigenous harvest and other socioecological practices early in the project may

allow for quickly gaining the cultural benefits of bison harvest while proactively mitigating adverse ecosystem impacts [83,124]. In contrast, Yellowstone has had free-roaming bison for over a century [63], and populations have reached a point where ecosystem recovery is challenged by overabundant bison, as Indigenous people remain excluded from harvesting or burning the park [19,23,25]. Without policy adjustments and management changes, both Banff and Yellowstone may risk ecological and sociological consequences of overabundant bison populations in historically low-abundance areas [23,79,83,121].

Adjusting 19th- and 20th-century socio-political decisions that excluded Indigenous people [19,20] and restoring their socioecological role in managing both bison and their ecosystems may have numerous benefits [6]. In addition, the human harvesting of bison is only one process in complex Cordilleran ecosystems and cannot be restored in isolation—it must be done in the context of a holistic ecosystem restoration method that recognizes long-term human roles as not only harvesters but also as gatherers, cultivators, and prescribed burners [30,88,113,114,125–128]. There are several examples of successful co-management models built on strong relationships between science and traditional ecological knowledge [6,129,130], and we encourage bison restoration projects to follow suit. Co-managing bison with Indigenous peoples and restoring the top-down harvest pressure that they provide may be the key to restoring bison and their full range of ecological and cultural benefits to the Western Cordillera and across North America.

**Supplementary Materials:** The supporting information can be viewed at: https://lensoftimenorthwest.com/themes/lens-northwest-files/google-earth-map-journal-wildlife-observations/ (accessed on 26 October 2021).

**Author Contributions:** Conceptualization, C.A.W.; methodology, C.A.W. and J.J.F.; software, C.A.W. and J.J.F.; validation, C.A.W. and J.J.F.; formal analysis, J.J.F.; investigation, C.A.W. and J.J.F.; resources, C.A.W. and J.J.F.; data curation, C.A.W.; writing—original draft preparation, C.A.W. and J.J.F.; writing—review and editing, J.J.F. and C.A.W.; visualization, J.J.F.; supervision, C.A.W.; project administration, C.A.W. and J.J.F.; funding acquisition, C.A.W. All authors have read and agreed to the published version of the manuscript.

**Funding:** Parks Canada provided initial funding for the historical journal database compilation.

**Data Availability Statement:** Geospatial data and R code supporting the reported results of this study are openly available on GitHub at https://github.com/jfarr99/FarrandWhite_BuffaloOnTheEdge_November2022 (accessed on 29 October 2022). The full dataset of historical journal observations is available at https://lensoftimenorthwest.com/themes/lens-northwest-files/google-earth-map-journal-wildlife-observations/ (accessed on 29 October 2022) or after 2025 in the Whyte Museum of the Canadian Rockies, Banff, Alberta, Clifford A. White fonds.

**Acknowledgments:** We appreciate the comments from M. Hebblewhite, K. Heuer, and C. Lewis on earlier drafts of this manuscript, as well as those from reviewers T. Binnema and C. Kay suggested sources for historical journal information. We respectfully acknowledge that this work was conducted on Treaty 7 territory, a traditional gathering place for diverse Indigenous peoples, including the Blackfoot Confederacy (comprising the Siksika, Piikani, and Kainai First Nations), as well as the Tsuut'ina First Nation, and the Stoney Nakoda (including the Chiniki, Bearspaw, and Wesley First Nations). We hope that our future work on bison will be conducted in collaboration with these peoples.

**Conflicts of Interest:** The authors declare no conflict of interest.

**Appendix A**

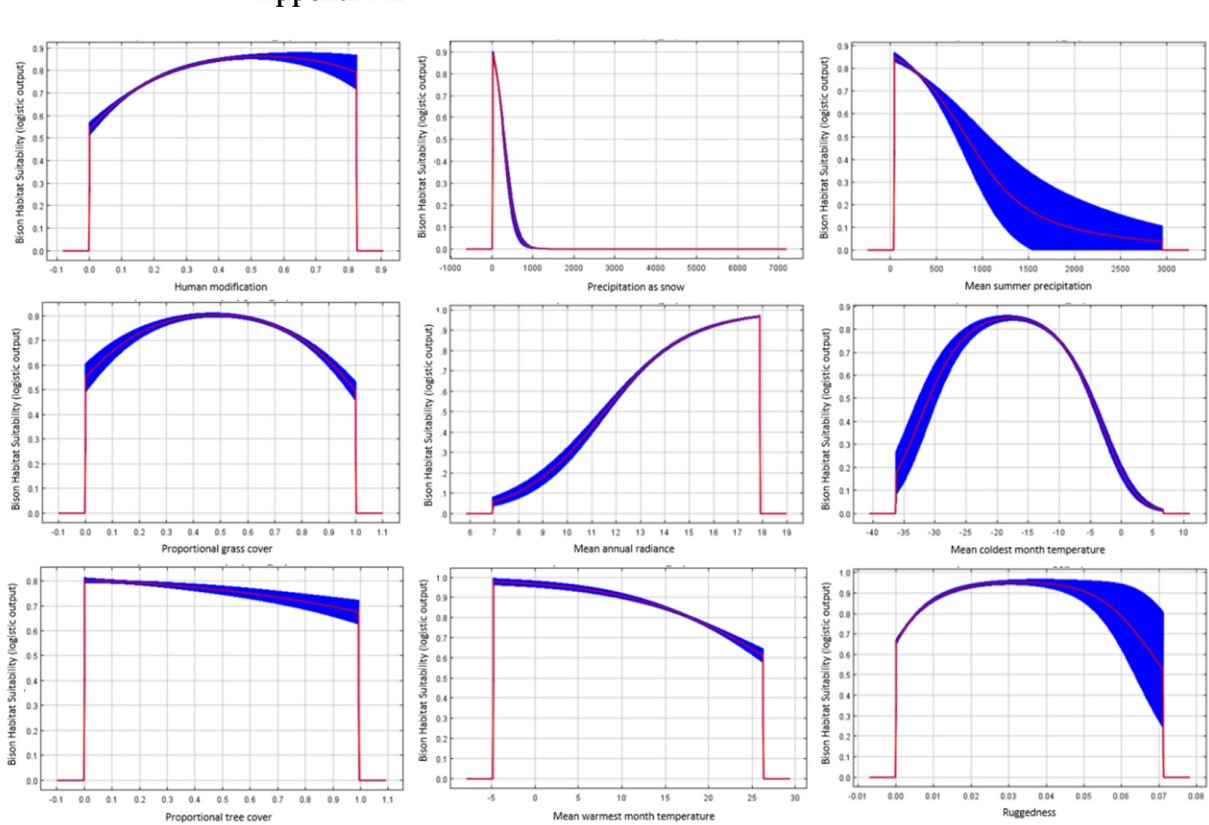

**Figure A1.** Marginal response curves revealing the relationship between predictor variables (*x* axes) and bison habitat suitability (*y* axes) as determined by the top selected MaxEnt model. Red lines show mean response of 10 replicate MaxEnt runs and blue shaded area shows the standard deviation.

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
