# Peer review of "Buffalo on the Edge: Factors Affecting Historical Distribution and Restoration of Bison bison in the Western Cordillera, North America"

_diversity, doi:10.3390/d14110937_

Round 1

Reviewer 1 Report

This manuscript evaluates environmental and human pressures that influenced the historical distribution of bison in the Western Cordillera using historical journal records, and suggests how these factors should be considered for future restoration efforts of bison in context of both location and management, specifically indigenous approaches. The authors found that the most influential factor was suitable habitat though this did not account for rare sightings of bison in some of the western portions of the range. It is likely that human pressures from harvesting created limited sink populations in these areas. The authors propose that without adding back the human pressures, namely harvesting of bison by Indigenous people, that restoration efforts could result in damage to the ecosystem. This paper will be of interest to a wide variety of audiences interested in bison restoration in the North American west and provides data that should be considered when deciding where to establish new bison herds and how they could be managed.

General comments

I can see in figure 2A that journal observations were made across the range though they were more frequent in the southern portions of the study area. Are the authors concerned that external factors such as climate conditions impacted the presence of humans in certain areas and thus the records of species abundance in those areas? How did the authors account for this?

Starting line 154, the temporal span of the land cover, topo data, and climatic predictors from the ArcGIS software is unclear to me. If it doesn’t go back to when bison were observed, how far does it go back? Or did you use current day estimates for these parameters or those estimated as of 2005 (line 161)?

Line 315, the proposal is to use Indigenous harvest in bison sink areas similar to what occurred in the 1700s and 1800s. While the cultural value of this approach is clear to me, the paper does not clearly explain how indigenous harvesting methods are more advantageous than other methods of harvesting, for example, managing a population by hunting via sale of tags. An additional comment: the way it is presented, this approach is primarily advantageous for ecosystem restoration and cultural reconnection and secondarily to bison restoration as a species (e.g. in terms of managing herds for genetics or maintenance of family groups/multiple generations on the landscape), though the two certainly go hand-in-hand. Perhaps this was the intent of the authors.

Specific Comments

Line 66 – remove “to” before “east”

Line 81 – change “has” to “have”

Line 83 – suggest adding that the humans, bears, and wolves are the primary predators of bison in this sentence.

Line 110 – Rather than “For humans...” suggest “For determining human abundance, ...”

Figure 2, the legend for the abundance indices was confusing. As formatted, it appeared that bison observations were shown in panel A, humans in B, etc until I reached the end and realized that they were shifted. I suggest creating some space between the color indices and the next species listed. Also, there is a Euro sign used for E in the legend.

Line 385, change “the” to “they”

Line 439, appears to have the remnant of a sentence left behind. Remove “Working with Indigenous groups that have deep cultural connections”

Reviewer 2 Report

This paper explores the relative impact of bottom up (climate, land cover, topography) and top down (human and non-human predation) on the distribution of bison in the intermountain west between AD 1691 and 1928. The authors approach this problem through the analysis of >9000 historical records, gleaned from the literature. The topic and approach are novel. I applaud the authors for thinking outside of the box to get at what may be important regional aspects of bison behavior and landscape use. They also use insights gained through the study to comment on modern conservation and bison management issues in the intermountain west–which is also a good thing. That said…some of their analyses and subsequent interpretations of results are lacking in historical (or archaeological) context, leading to over-statement of and over-confidence in their results. This is especially important because they state that pre-Euroamerican indigenous groups in the intermountain west exerted strong population controls on bison through harvest pressure. 

1. Figure 1. Although this is a classic figure by Allen, it should be redrafted for the current paper. The legend is unreadable and details of different bison ranges are not clear. 

Lines 60-76. The potential impacts of people and human communities are a very important part of this study. Furthermore, one of the key goals seems to be to evaluate the impact of indigenous groups on the distribution of bison in the IMW. However, this background neglects to include ANY mention of the rich archaeological and ethnohistoric record for the region. Key papers on Holocene bison distribution are not included (see a number of papers by Don Grayson and Karen Lupo). Nor does it provide a background on regional climate trends during the historic period (Little Ice Age is ~AD 1600-1850) and their impact on regional vegetation. 

As written/conceived, this study over-simplifies the potential drivers in bison distribution throughout the IMW during this time period. 

Line 155. The period of interest corresponds to the Little Ice Age climate anomaly, where northern hemisphere climates were noticeably cooler than the late Holocene and historic average. Using modern/historic climate data is not appropriate since data covering the LIA are out there. For instance, Tree ring derived Palmer Drought Severity Index (Cook et al., 2004: https://www.ncei.noaa.gov/access/paleo-search/study/6232) provides information for this time period, as does Kaufmann et al. 2020 (https://www.ncei.noaa.gov/access/paleo-search/study/27330). Paleoclimate archives are not as highly resolved as historical data, but they exist and are more appropriate, especially since the period of interest corresponds to a known swing towards cooler climates (Little Ice Age). 

Treating the period 1600-1920 categorically, with assumptions of stasis in human populations and climate is problematic. There are time-transgressive trends in the region that had profound impacts on the distribution of bison. Euroamerican settlement, industrial-scale mining, European introduced diseases (zoonoses + human infections), and market hunting all had a major impact on the distribution of wildlife populations. In light of these major changes occurring in the region, this analysis seems reductionist. Historical context must be taken into account, either by tempering interpretations in the discussion section, or by explicitly taking some of these historical processes into account in the analysis itself. 

Figure 3. What do bison habitat suitability values mean? Please make this clear in the legend– 1=poor? 10=good?

The discussion touches on some good points, but it overstates the results in a couple of significant ways. 

Lines 299-301. The conceptualization of IMW bison as a sink population is incorrect. On the basis of archaeological (late Holocene) data, bison are endemic to the region but never at the density levels we might find in the ‘core’ region of the Great Plains. Although some animals may be dispersing through mountain corridors from east of the Rockies to get there, it is more likely that local herds of bison are increasing or decreasing according to a) short term climate conditions, and/or b) changes in predation (human or non-human). I’m not sure what is intended by the concept of an ‘ecological trap’ for bison referred to in line 310, but I’m pretty sure that it doesn’t have much to do with connectivity to the Great Plains core. 

Lines 325-327. The differences between Plains and Wood bison are not taxonomically significant (see Shapiro et al., 2004, Ball et al., 2015; Geist 1991; Van Zyll de Jong, 1986) and sets up a false dichotomy of landscape use between the two ‘groups’. 

Lines 339-343. Intertribal buffer zones. This is an interesting concept, but keep in mind that its social origins are specific to a certain place and time and may not be broadly applicable to the inter-mountain west due to low human population densities. Also…these zones have their origins in the competition for buffalo robe trade, so are unlikely to apply deeper into time. 

Lines 372 and 373. The extirpation of bison <3000 BP and the development of the hide trade routes (~ 200 BP) are separated by potentially 2800 years. Stating a causal relationship between the two on the basis of the current study is a big stretch.

Reviewer 3 Report

The manuscript presented by the authors analyse the historical dispersion of Bison in Western North America. This paper is based on a collect of data from historical sources pre-dating or contemporaneous of the great near-extinction of American bisons. This approach is very interesting and give insight about Human-bison interaction before the intensification of the colonisation and ecological preferences of bison. In the end, it provides conclusion about present-day bison herd management.

Overall, this work is very interesting, well-written and well-explain and should be published.

However, in my opinion, a few numbers of points should be clarified in order to improve the manuscript.

All my comments are listed below:

Introduction

l. 82- 87 : The authors explain directly that they will consider the distribution of bison and 3 great predators: humans, bear and wolf. Could just before the author detail a bit what are the main bison predators and explain why, according to them, bear, wolves and humans have a greater impact than other predators like puma for example?

Method

l. 99-100: the authors used wildlife observation from a period from 1691-1860 in southern Canada and Northern USA and 1770-1928 for northern Canada and eastern Alaska. Thus, for the south region of the study, they had access to record pre-near-extinction of American bison while for the northern region they are studying, there data post-date this period. Could they develop a bit more why they think those data are comparable between them to understand Bison repartition pre-near-extinction? More precisely, how the northern data can be representative of Bison historical distribution?

Discussion

L. 383-386: the authors state that female perspective can give inputs about the seasonal availability of resources. Could the authors explain better that aspect of historical biases in their sources? It would be great to have some detail about ethnological/anthropological indigenous tribes gender organization regarding this issue! More detail and references should be added up here.

l. 399-401: The authors discuss a bit the historical distribution of Bison on a larger time-scale. They could also cite work from zooarchaeologist or palaeontologist describing Bison in various location of North America as South as Mexico.

See for example ; Eduardo Jiménez-Hidalgo, Lucía Cabrera-Pérez, Bruce J. MacFadden, Rosalía Guerrero-Arenas, First record of Bison antiquus from the Late Pleistocene of southern Mexico, Journal of South American Earth Sciences, Volume 42, 2013, Pages 83-90, https://doi.org/10.1016/j.jsames.2012.07.011.

This information will back up their observation of a very flexible animal which had inhabited a large range of biotope.

l. 445: avoid using immemorial. It is “since the Late Pleistocene”.

Reviewer 4 Report

First I would like to thank the authors for contributing their work to the scientific literature. I found your work interesting. I do however have some concerns about your rather circular analysis.

Primaarily, why didn't you just incorporate your top-down factors into your MaxEnt analysis? The distribution of bison was historically affected by predators, and predator presence may have been influenced by environmental variables creating a web of interactions and positive and negative impacts. I don't think just removing predators from your MaxEnt analysis gives you an unbiased estimation of habitat suitability, that you can then reconnect with predator distributions. You have effectively limited the power of your environmental variables in predicting the distribution of bison.

Secondly, I think you need to include more land use variables in your MaxEnt analysis. A measure of modern human land use (eg buildings, clear fell logging, and changes to wetlands, swamps or rivers must be incorporated) if you are going to use MODIS 2005 data. Many cities are more than 10x10km. What habitat were those areas in the past?

Thirdly, the distribution of journal observations covers a very regular almost square area of the NW. Please state any boundaries to your collection of journals and observations. Bison abundance appears to be high in areas south of your study area and this may influence your MaxEnt model. The results of your best MaxEnt model make it sound like bison are a southern species preferring warm grasslands. Evidently you need to include data from SW areas of the USA in your MaxEnt model.

Your discussion ignores the socio-political influences on over-harvest. Why would over-harvest occur? Humans use more resources than they need these days because we sell them/trade them with populations that do not have direct access to the resources. Further, hunting pre-mechanised firearms required getting close to very large animals and was inherently dangerous. Why would humans risk their life to harvest a product with relatively limited trading potential? Europeans had mechanised firearms and trade routes with other continents, so the benefits to over-harvest were completely different. These issues should be discussed.

The discussion is not particularly well thought out and contains several broad sweeping conclusions that are not supported by the data. This needs to be reviewed. For example, reinstating a cultural practice can not recreate source-sink population dynamics for a species that does not have access to effective dispersal corridors.

I have also made other minor text edits and requests on the attached pdf.

Despite these issues, which I think can be remedied, the work is interesting and worth publishing.

Round 2

Reviewer 2 Report

Authors have address my previous concerns.